# MODEL-BASED OPPONENT MODELING

## ABSTRACT

When one agent interacts with a multi-agent environment, it is challenging to deal with various opponents unseen before. Modeling the behaviors, goals, or beliefs of opponents could help the agent adjust its policy to adapt to different opponents. In addition, it is also important to consider opponents who are learning simultaneously or capable of reasoning. However, existing work usually tackles only one of the aforementioned types of opponent. In this paper, we propose *model-based opponent modeling* (MBOM), which employs the environment model to adapt to all kinds of opponent. MBOM simulates the recursive reasoning process in the environment model and imagines a set of improving opponent policies. To effectively and accurately represent the opponent policy, MBOM further mixes the imagined opponent policies according to the similarity with the real behaviors of opponents. Empirically, we show that MBOM achieves more effective adaptation than existing methods in competitive and cooperative environments, respectively with different types of opponents, *i.e.*, fixed policy, naïve learner, and reasoning learner.

## 1 INTRODUCTION

Reinforcement learning (RL) has made great progress in multi-agent competitive games, *e.g.*, AlphaGo (Silver et al., 2016), OpenAI Five (OpenAI, 2018), and AlphaStar (Vinyals et al., 2019). In multi-agent environments, an agent usually has to compete against or cooperate with diverse other agents (collectively termed as *opponents* whether collaborators or competitors) unseen before. Since the opponent policy influences the transition dynamics experienced by the agent, interacting with diverse opponents makes the environment seem nonstationary from the agent's perspective. Due to the complexity and diversity in opponent policies, it is very challenging for the agent to retain overall supremacy.

Explicitly modeling the behaviors, goals, or beliefs of opponents (Albrecht & Stone, 2018), rather than treating them as a part of the environment, could help the agent adjust its policy to adapt to different opponents. Many studies rely on predicting the actions (He et al., 2016; Hong et al., 2018; Grover et al., 2018; Papoudakis & Albrecht, 2020) and goals (Rabinowitz et al., 2018; Raileanu et al., 2018) of opponents during training. When facing diverse or unseen opponents, the agent policy conditions on the prediction or representation generated by corresponding modules. However, opponents may also have the same reasoning ability, *e.g.*, an opponent who makes prediction about the agent's goal. In this scenario, higher-level reasoning and some other modeling techniques are required to handle such sophisticated opponents (Wen et al., 2019; Yang et al., 2019; Yuan et al., 2020). In addition, the opponents may learn simultaneously, and the modeling becomes unstable and the fitted models with historical experiences lag behind. To enable the agent to continuously adapt to learning opponents, LOLA (Foerster et al., 2018a) takes into account the gradients of the opponent's learning for policy update, Meta-PG (Al-Shedivat et al., 2018) formulates continuous adaptation as a meta-learning problem, and Meta-MAPG (Kim et al., 2021a) combines meta-learning with LOLA. However, LOLA requires knowing the learning algorithm of opponents, while Meta-PG and Meta-MAPG require all opponents use a same learning algorithm.

Unlike existing work, we do not make such assumptions and focus on enabling the agent to learn effectively by directly representing the policy improvement of opponents when interacting with them, even if they may be also capable of reasoning. Inspired from the intuition that human could anticipate the future behaviors of opponents by simulating the interactions in the brain after knowing the rules and mechanics of the environment, in this paper, we propose *model-based opponent modeling*

(MBOM), which employs the environment model to predict and capture the policy improvement of opponents. By simulating the interactions in the environment model, we could obtain the best responses of opponents under the agent policy conditioned on the opponent model. Then, the opponent model can be fine-tuned using the simulated best responses to get the higher-level opponent model. The higher-level opponent model reasons more deeply and thus more competitive. By recursively repeating the simulation and fine-tuning, MBOM imagines the learning and reasoning of opponents and generates a set of opponent models with different levels, which could also be seen as recursive reasoning. However, since the learning and reasoning of opponents are *unknown*, a certain-level opponent model might overestimate or underestimate the opponent. To effectively and accurately represent the opponent policy, we further propose to mix the imagined opponent policies according to the similarity with the real behaviors of opponents updated by Bayesian rule.

We evaluate MBOM in the two competitive tasks, Triangle Game and One-on-One, against three types of opponent, *e.g.*, fixed policy, naïve learner, and reasoning learner. MBOM outperforms strong baselines, especially when against naïve learner and reasoning learner, which demonstrates the effectiveness of recursive imagination and Bayesian mixing. We also show the adaptation ability of MBOM in the cooperative task Coin Game.

## 2 RELATED WORK

### 2.1 OPPONENT MODELING

In multi-agent reinforcement learning (MARL), it is a big challenge to form robust policy due to the unknown opponent policy. From the perspective of an agent, if opponents are considered as a part of the environment, the environment is unstable and complex for policy learning when the policies of opponents are also changing. If the information about opponents is included, *e.g.*, behaviors, goals and beliefs, the environment may become stable, and the agent could learn using single-agent RL methods. This line of research is *opponent modeling*.

One simple idea of opponent modeling is to build a model each time a new opponent or group of opponents is encountered (Zheng et al., 2018). However, learning a model every time is not efficient. A more computationally tractable approach is to represent an opponent's policy with an embedding vector. Grover et al. (2018) uses a neural network as encoder, taking as input the trajectory of one agent. Imitation learning and contrastive learning are used to train the encoder. Then, the learned encoder can be combined with RL by feeding the generated representation into policy or/and value network. Learning of the model can also be performed simultaneously with RL, as an auxiliary task (Jaderberg et al., 2016). Based on DQN (Mnih et al., 2015), DRON (He et al., 2016) and DPIQN (Hong et al., 2018) use a secondary network which takes observations as input and predicts opponents' actions. The hidden layer of this network is used by the DQN module to condition on for a better policy. It is also feasible to model opponents using variational autoencoders (Papoudakis & Albrecht, 2020), which means the generated representations are no longer deterministic vectors, but high-dimensional distributions. ToMnet (Rabinowitz et al., 2018) tries to make agents have the same Theory of Mind (Premack & Woodruff, 1978) as humans. ToMnet consists of three networks, reasoning about the agent's action and goal based on past and current information. SOM (Raileanu et al., 2018) implements Theory of Mind from a different perspective. SOM uses own policy, opponent's observation and opponent's action to work backwards to learning opponent's goal by gradient ascent.

The methods aforementioned only consider opponent policies which are independent of the agent. If opponents hold belief about the agent, the agent can form higher level belief about opponents' belief. This process can perform recursively, termed *recursive reasoning*. PR2 (Wen et al., 2019) uses SAC (Haarnoja et al., 2018) to approximate opponents' policies conditioning on the agent's action. Since PR2 uses the agent's Q-function to estimate opponents' policies, it can only be applied in cooperative environments. Yuan et al. (2020) takes both level-0 and level-1 beliefs as input to the value function. Level-0 belief is updated according to Bayesian rule, and level-1 belief is updated using a learnable neural network. However, these methods use centralized training decentralized execution algorithms to train a set of fixed agents that cannot handle diverse opponents in execution. Bayes-ToMoP (Yang et al., 2019) is developed to detect and handle different reasoning opponents, but the implementation is complex and not scalable.

If the opponents are also learning, the modeling mentioned above becomes unstable and the fitted models with historical experiences lag behind. So it is beneficial to take into consideration the learning process of opponents. LOLA (Foerster et al., 2018a) introduces the impact of the agent's policy on the anticipated parameter update of the opponent. A neural network is used to model the opponent's policy and estimate learning gradient of the opponent's policy, implying that the learning algorithm used by the opponent should be known, otherwise the estimated gradient will be inaccurate. Further, the opponents may still be learning continuously during execution. Meta-PG (Al-Shedivat et al., 2018) is a method based on meta policy gradient, using trajectories come from the current opponents to do multiple meta-gradient steps and construct a policy that is good for the updated opponents. Meta-MAPG (Kim et al., 2021a) extends this method by including an additional term that accounts for the impact of the agent's current policy on the future policies of opponents, similar to LOLA. These meta-learning based methods require that the distribution of trajectories match across training and test, which implicitly means all opponent use a same learning algorithm.

## 2.2 MODEL-BASED RL AND MARL

Model-based RL allows the agent to have access to the transition function. There are two typical branches of model-based RL approaches: background planning and decision-time planning. In background planning, the agent could use the learned model to generate additional experiences for assisting learning. For example, Dyna-style algorithms (Sutton, 1990; Kurutach et al., 2018; Luo et al., 2019) perform policy optimization on simulated experiences, and model-augmented value expansion algorithms (Feinberg et al., 2018; Oh et al., 2017; Buckman et al., 2018) use model-based rollouts to improve the update targets. In decision-time planning (Chua et al., 2018), the agent could use the model to rollout the optimal action at a given state by looking forward during execution, *e.g.*, model predictive control.

The model-based MARL methods could be divided into two classes: centralized model and decentralized model. AMLAPN (Park et al., 2019) builds a centralized auxiliary prediction network to model the environment dynamics and the opponent actions to alleviate the non-stationary dynamics. A centralized multi-step generative model (Krupnik et al., 2020) with a disentangled variational auto-encoder has been proposed for performing trajectory planning. For decentralized model, IS (Kim et al., 2021b) uses the model to generate imagined trajectories for multi-agent communication to share intention. HPP (Wang et al., 2020) uses the models to propose and evaluate navigation subgoals for the rendezvous task.

*Our method relaxes the limitations on opponent modeling. We make no assumptions about the variation of opponents' policies. They could be fixed, randomly sampled from an unknowable policy set, or continuously learning using an unknowable and changeable RL algorithm, both in training and execution. We learn an environment model that allows the agent to perform recursive reasoning against opponents who may also have the same reasoning ability.*

## 3 PRELIMINARIES

We consider an $n$-agent stochastic game $(\mathcal{S}, \mathcal{A}^1, \ldots, \mathcal{A}^n, \mathcal{P}, \mathcal{R}^1, \ldots, \mathcal{R}^n, \gamma)$, where $\mathcal{S}$ is the state space, $\mathcal{A}^i$ is the action space of agent $i \in [1, \ldots, n]$, $\mathcal{A} = \prod_{i=1}^{n} \mathcal{A}^i$ is the joint action space of agents, $\mathcal{P} : \mathcal{S} \times \mathcal{A} \times \mathcal{S} \rightarrow [0, 1]$ is a transition function, $\mathcal{R}^i : \mathcal{S} \times \mathcal{A} \times \mathcal{S} \rightarrow \mathbb{R}$ is the reward function of agent $i$ , and $\gamma$ is the discount factor. The policy of agent $i$ is $\pi^i$, and the joint policy of other agents is $\pi^{-i}(a^{-i}|s) = \prod_{j \neq i} \pi^j(a^j|s)$, where $a^{-i}$ is the joint action except agent $i$. All agents interact with the environment simultaneously without communication. The historical trajectory is available, *i.e.*, for agent $i$ at timestep $t$, $\{s_0, a_0^i, a_0^{-i}, \ldots, s_{t-1}, a_{t-1}^i, a_{t-1}^{-i}\}$ is observable. The goal of the agent $i$ is to maximize its expected cumulative discount rewards

$$\mathbb{E}_{\substack{s_{t+1} \sim \mathcal{P}(\cdot|s_t, a_t^i, a_t^{-i}), \\ a^i \sim \pi^i(\cdot|s_t), a_t^{-i} \sim \pi^{-i}(\cdot|s_t)}} \left[ \sum_{t=0}^{\infty} \gamma^t \mathcal{R}^i(s_t, a_t^i, a_t^{-i}, s_{t+1}) \right]. \tag{1}$$

For convenience, the learning agent treats all other agents as a joint opponent with the joint action $a^o \sim \pi^o(\cdot|s)$ and reward $r^o$. The action and reward of the learning agent are denoted as $a \sim \pi(\cdot|s)$ and $r$, respectively.

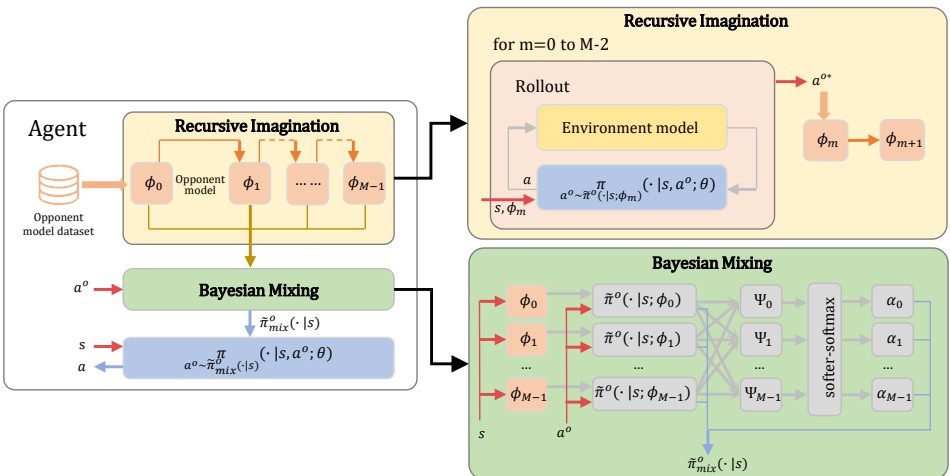

Figure 1: Architectures of MBOM

# 4 MODEL-BASED OPPONENT MODELING

MBOM employs the environment model to predict and capture the learning of opponent policy. By simulating recursive reasoning via the environment model, MBOM imagines the learning and reasoning of the opponent and generates a set of opponent models. To obtain stronger representation ability and accurately capture the adaptation of the opponent, MBOM mixes the imagined opponent policies according to the similarity with the real behaviors of opponent.

## 4.1 RECURSIVE IMAGINATION

If the opponent is also learning during interaction, the opponent model fitted with historical experiences always lag behind, making the agent hard to adapt to the opponent. Moreover, if the opponent could adjust its policy according to the actions, intentions, or goals of the agent, then recursive reasoning may occur between agent and opponent. However, based on the lagged opponent model, the agent would struggle to keep up with the learning of opponent. To adapt to the learning and reasoning opponent, the agent should predict the current opponent policy and reason more deeply than the opponent.

MBOM explicitly simulates the recursive reasoning process utilizing the environment model, called *recursive imagination*, to generate a series of opponent models, called Imagined Opponent Policies (IOPs). Initially, the agent interacts with $\eta$ different opponents which are learning with PPO (Schulman et al., 2017), and collect a buffer $\mathcal{D}$ which contains the experience $\langle s, a, a^o, s', \boldsymbol{r} \rangle$, where $\boldsymbol{r} = \langle r, r^o \rangle$. For zero-sum game $(r + r^o = 0)$ and fully cooperative game $(r = r^o)$, $r^o$ can be easily obtained, while for general-sum game we make a mild assumption that $r^o$ can be accessed during experience collection. Using the experience buffer $\mathcal{D}$, we can train the the environment model $\Gamma(s', \boldsymbol{r}|s, a, a^o; \zeta)$ by minimizing the mean square error

$$\mathbb{E}_{s,a,a^o,s',\boldsymbol{r}\sim\mathcal{D}} \frac{1}{2}\|s' - \hat{s}'\|^2 + \frac{1}{2}\|\boldsymbol{r} - \hat{\boldsymbol{r}}\|^2, \tag{2}$$

where $\hat{s}', \hat{\boldsymbol{r}} = \Gamma(s, a, a^o; \zeta)$, obtain the level-0 IOP $\tilde{\pi}^o(\cdot|s; \phi_0)$ by maximum-likelihood estimation

$$\mathbb{E}_{s,a^o\sim\mathcal{D}} \log \tilde{\pi}^o(a^o|s; \phi_0), \tag{3}$$

and learn the policy of the agent $\pi(\cdot|s, a^o; \theta)$ using PPO. By running PPO, we could also get the value function $V(s)$ of the agent. To imagine the learning of the opponent, as shown in Figure 1, we use the rollout algorithm (Tesauro & Galperin, 1996) to get the best response of the opponent under the agent policy $\pi$. For each opponent action $a^o_t$ at timestep $t$, we uniformly sample the opponent action sequences in the next $k$ timesteps, simulate the trajectories using the learned environment

---

**Algorithm 1** MBOM

---

1: **Pretraining:**
2: Initialize the recursive imagination layer $M$.
3: Initialize the weights $\boldsymbol{\alpha}$ with $(1, 0, \ldots, 0)$.
4: Interact with $\eta$ learning opponents and collect the experience buffer $\mathcal{D}$.
5: Train the level-0 IOP $\phi_0$, the environment model $\Gamma_\zeta$, and the agent policy $\theta$ using the experience buffer $\mathcal{D}$.
6: **Interaction:**
7: **for** $t = 1, \ldots,$ max_epoch **do**
8:     Interact with the opponent to observe the real opponent actions.
        {*//Recursive Imagination*}
9:     Fine-tune $\phi_o$ with the real opponent actions
10:    **for** $i = 1, \ldots, M - 1$ **do**
11:        Rollout in $\Gamma_\zeta$ with $\pi(\cdot|s, a^o \sim \tilde{\pi}^o(\cdot|s; \phi_{i-1}); \theta)$ to get the best responses of the opponent $a^{o*}$ by equation 4 or equation 5.
12:        Fine-tune $\phi_{i-1}$ with the best responses to get the $\phi_i$.
13:    **end for**
        {*//Bayesian Mixing*}
14:    Update $\boldsymbol{\alpha}$ by equation 7.
15:    Mix the IOPs to get $\tilde{\pi}^o_{\text{mix}}$ by equation 6.
16: **end for**

---

model $\Gamma_\zeta$, and select the best response with the highest rollout value

$$a_t^{o*} = \underset{a_t^o}{\operatorname{argmax}} \max_{a_{t+1}^o, \cdots, a_{t+k}^o \sim \text{Unif}} \sum_{j=0}^{k} \gamma^{t+j} r_{t+j}^o. \tag{4}$$

During the simulation, the agent acts according to the policy conditioned on the modeled opponent policy, $a_t \sim \pi(\cdot|s_t, a_t^o \sim \tilde{\pi}^o(\cdot|s_t; \phi_0); \theta)$, and the learned environment model provides the transition $s_{t+1}, \boldsymbol{r_t} = \Gamma(s_t, a_t, a_t^o; \zeta)$. With larger $k$, the rollout has longer planning horizon, and thus could evaluate the action $a^{o*}$ more accurately, assuming a perfect environmental model. However, the computation cost of rollout increases exponentially with the planning horizon to get an accurate estimate of $a^{o*}$, while in practice the compounding error of the environmental model also increases with the planning horizon. Therefore, the choice of $k$ is a trade-off between accuracy and cost. Specifically, for zero-sum game and fully cooperative game, we can approximately estimate the opponent state value $V^o(s)$ as $-V(s)$ and $V(s)$, respectively, and modify the rollout value like $n$-step return (Sutton & Barto, 2018) to obtain a longer horizon

$$a_t^{o*} = \underset{a_t^o}{\operatorname{argmax}} \max_{a_{t+1}^o, \cdots, a_{t+k}^o \sim \text{Unif}} \sum_{j=0}^{k} \gamma^{t+j} r_{t+j}^o + \gamma^{t+k+1} V^o(s_{t+k+1}). \tag{5}$$

By imagination, we can obtain the best response of the opponent under the agent policy $\pi$ and level-0 IOP $\tilde{\pi}^o(\cdot|s; \phi_0)$ and construct the simulated data $\{\langle s, a^{o*}\rangle\}$. Then, we use the data to fine-tune the level-0 IOP $\tilde{\pi}^o(\cdot|s; \phi_0)$ by maximum-likelihood estimation, and obtain the level-1 IOP $\tilde{\pi}^o(\cdot|s; \phi_1)$. The level-1 IOP can be see as the best *response* of agent to the *response* of opponent to level-0 IOP. In the imagination, it is the nested form as "the opponent believes [that the agent believes (that the opponent believes ...)]." The existing imagined opponent policy is the innermost "(that the opponent believes)," while the outermost "the opponent believes" is $a^{o*}$ which is obtained by rollout process. Recursively repeating the rollout and fine-tuning, where the agent policy is conditioned on the IOP $\tilde{\pi}^o(\cdot|s; \phi_{m-1})$, we could derive the level-$m$ IOP $\tilde{\pi}^o(\cdot|s; \phi_m)$. The higher-level IOP reasons more deeply, and thus more competitive.

## 4.2 Bayesian Mixing

By recursive imagination, we get $M$ IOPs with different reasoning levels. However, since the learning and reasoning of the opponent are unknown, a single IOP might overestimate or underestimate the opponent. To obtain stronger representation ability and accurately capture the learning of the opponent, as illustrated in Figure 1, we linearly combine the IOPs to get a mixed policy, similar idea

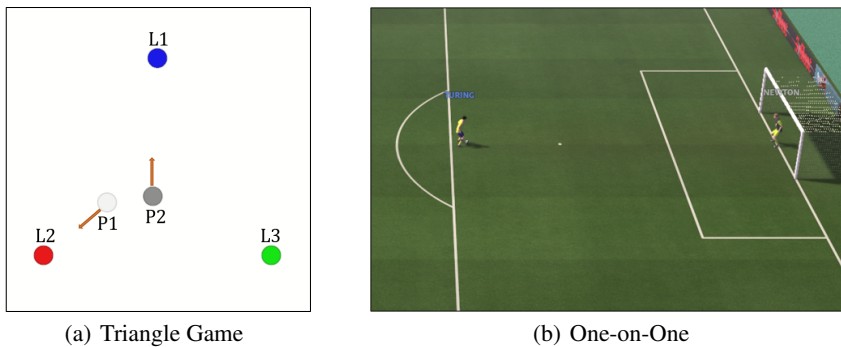

(a) Triangle Game                               (b) One-on-One

Figure 2: Illustrations of the scenarios.

was also present in (Wen et al., 2020)

$$\tilde{\pi}_{\text{mix}}^{o}(\cdot|s) = \sum_{i=0}^{M-1} \alpha_i \tilde{\pi}^o(\cdot|s; \phi_i), \tag{6}$$

where $\alpha_i$ is the weight of level-$i$ IOP, which is produced by the IOPs mixer,

$$\boldsymbol{\alpha} = (\alpha_0, \ldots, \alpha_{M-1}) = \text{softer-softmax}(\Psi_0, \ldots, \Psi_{M-1}). \tag{7}$$

Softer-softmax (Hinton et al., 2015) is a variant of softmax, which uses higher temperature to control a softy of the probability distribution over classes. $\Psi_m$ is the decayed moving average of $p(m|a^o)$, which is the probability of using the level-$m$ IOP given the action of the opponent $a^o$. By Bayesian rule, we have

$$p(m|a^o) = \frac{p(a^o|m)p(m)}{\sum_{i=0}^{M-1} p(a^o|i)p(i)} = \frac{\tilde{\pi}^o(a^o|s; \phi_m)p(m)}{\sum_{i=0}^{M-1} [\tilde{\pi}^o(a^o|s; \phi_i)p(i)]}, \tag{8}$$

where $p(m)$ is the probability of using the level-$m$ IOP, and we estimate it as the moving average of $p(m|a^o)$. $\Psi_m$ indicates the similarity between the level-$m$ IOP and the opponent in the most recent stage. Given the opponent actions, the higher $\Psi_m$ means that the actions are more likely to be generated from the level-$m$ IOP, and thus the level-$m$ IOP is more similar to the opponent. Adjusting the weights $\boldsymbol{\alpha}$ according to the similarity could obtain a more accurate estimate of the improving opponent policy. Moreover, the IOPs mixer is non-parametric, which could be updated quickly and efficiently without parameter training and too many interactions. Therefore, the IOPs mixer could adapt to the fast-improving opponent.

For completeness, the whole procedure of MBOM in given in Algorithm 1.

## 5 EXPERIMENTS

### 5.1 SETTING UP

We evaluate MBOM in two competitive environments.

**Triangle Game** is an asymmetric zero-sum game on two-dimension with multi-step actions, implemented on Multi-Agent Particle Environment (Mordatch & Abbeel, 2017; Lowe et al., 2017) (MIT license). As shown in Figure 2(a), there are two moving players, P1 and P2, and three fixed landmarks, L1-L3, in a square field. The landmarks are located at the three vertexes of an equilateral triangle with the side length 0.6. When the distance between a player and a landmark is less than 0.15, the agent touches the landmark and has the state T. T1 indicates that the player touches the landmark L1, and so on. If the player does not touch any landmark, the player state is F. The payoff matrix of the two players is shown in Table 2(a). P2 has inherent disadvantages since the optimal solution of P2 always strictly depends on the state of P1. When facing different policies of P1, P2 has to adjust its policy to adapt to P1 for higher reward. We control P2 as the agent and take P1 as the opponent.

**One-on-One** is a two-player competitive game implemented on Google Research Football Environment (Kurach et al., 2020) (Apache-2.0 License), a physics-based 3D simulator. As shown in

Table 1: Payoff matrix of Triangle Game.

|  |  | Player 2 | | | |
|---|---|---|---|---|---|
|  |  | F | T1 | T2 | T3 |
| Player1 | F | $0/0$ | $-0.5/+0.5$ | $-0.5/+0.5$ | $-0.5/+0.5$ |
|  | T1 | $+0.5/-0.5$ | $+1/-1$ | $+1/-1$ | $-1/+1$ |
|  | T2 | $+0.5/-0.5$ | $-1/+1$ | $+1/-1$ | $+1/-1$ |
|  | T3 | $+0.5/-0.5$ | $+1/-1$ | $-1/+1$ | $+1/-1$ |

Figure 2(b), there are two players, the shooter, which could dribble and shoot the ball, and the goal-keeper. The shooter controls the ball in the initial state. The episode ends after 30 timesteps or when the shooter loses possession of the ball. At the end of an episode, if the shooter shoots the ball into the goal, the shooter will get a reward $+1$, and the goalkeeper will get a reward $-1$. Otherwise, the shooter will get a reward of $-1$, and the goalkeeper will get a reward of $+1$. The goalkeeper could only passively react to the strategies of the shooter and makes policy adaptation when the shooter strategy changes. We control the goalkeeper as the agent and take the shooter as the opponent.

**Baselines.** In the experiments, we compare MBOM with the following methods:

- LOLA-DiCE (Foerster et al., 2018b) is an expansion of the LOLA, which uses Differentiable Monte-Carlo Estimator (DiCE) operation to considers how to shape the learning dynamics of other agents.
- Meta-PG (Al-Shedivat et al., 2018) uses trajectories come from the current opponents to do multiple meta-gradient steps and construct a policy that is good for the updated opponents.
- Meta-MAPG (Kim et al., 2021a) includes an additional term that accounts for the impact of the agent's current policy on the future policies of opponents, compared with Meta-PG.
- PPO (Schulman et al., 2017) is a classical single agent RL algorithm, without any other modules.
- MBOM w/o IOPs always uses $\phi_0$ as the opponent model, without recursive imagination and Bayesian mixing.

The baselines have the same neural network architectures as MBOM. All the models are trained for five runs with different random seeds. All the curves are plotted using mean and standard deviation. More details about experimental settings and hyperparameters are available in Appendix A.

**Opponents.** For the opponents, we run independent PPO (Schulman et al., 2017) algorithms for 10 times. During each running, we store 20 opponent policies in the training set, 3 opponent policies in the validation set, and 3 opponent policies in the test set. So the sizes of the training set, validation set, and test set are 200, 30, and 30. We pre-train the agent with the opponents in the training set and make the agent interact with the opponents in the test set to evaluate the ability of generalization and adaptation. The validation set is only required by Meta-PG and Meta-MAPG. We construct three types of opponents:

- Fixed policy. The opponents in the test set will not update during the interaction.
- Naïve learner. The opponents in the test set will be updated with PPO during the interaction.
- Reasoning learner. The opponents could model the behavior of the agent and will be fine-tuned during the interaction.

To increase the diversity of the opponents, we adopt the reward shaping technique and add invisible barriers in the environment during the training.

## 5.2 PERFORMANCE

The experimental results against test opponents are shown in Figure 3, and the mean performance with standard deviation over all test opponents is summarized in Table 2 and 3. In Triangle Game, the opponent could take different strategies, *e.g.*, hovering around a landmark, commuting between two landmarks, or rotating among three landmarks, whereas the agent has to anticipate and adapt to the opponent strategies. The learning of LOLA-DiCE depends on the gradient of the opponent

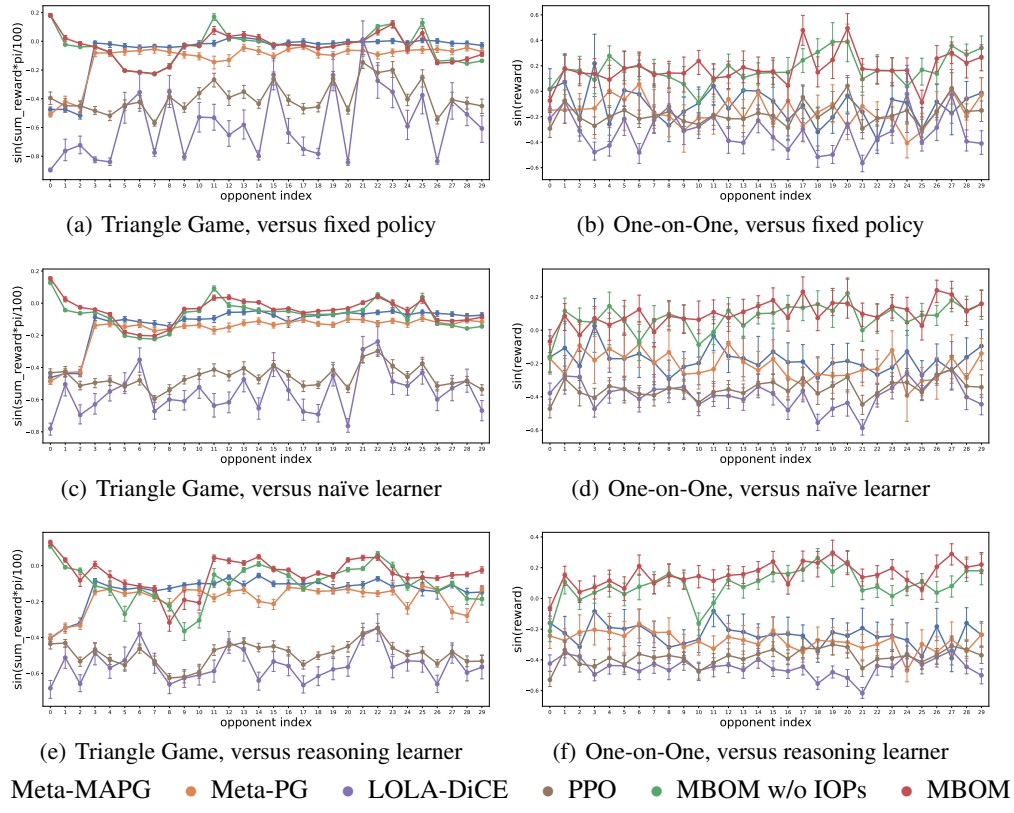

(a) Triangle Game, versus fixed policy      (b) One-on-One, versus fixed policy

(c) Triangle Game, versus naïve learner      (d) One-on-One, versus naïve learner

(e) Triangle Game, versus reasoning learner      (f) One-on-One, versus reasoning learner

● Meta-MAPG    ● Meta-PG    ● LOLA-DiCE    ● PPO    ● MBOM w/o IOPs    ● MBOM

Figure 3: Performance of adaptation against different types of opponents, *i.e.*, fixed policy, naïve learner, and reasoning learner. The results are plotted using mean and standard deviation with five different random seeds (*x*-axis is opponent index). The results show that MBOM can outperform other baselines, especially against naïve learner and reasoning learner.

model. However, the gradient information cannot clearly reflect the distinctions between diverse opponents, which leads to that LOLA-DiCE cannot adapt to the unseen opponent quickly and effectively. Meta-PG and Meta-MAPG show stronger adaptation abilities than LOLA-DiCE when facing different opponents, but fail on some opponents (index 0, 1, and 2 in Figure 3(a), 3(c), and 3(e)). By visualization, we find that those opponents are hovering around one landmark, which are out of the training set. Since Meta-PG and Meta-MAPG assume that the test set follows the distribution of the training set, these two methods are unable to adapt to the out-of-distribution opponents. MBOM w/o IOPs obtains similar results to MBOM when facing fixed policy opponents because $\phi_0$ could accurately predict the opponent behaviors if the opponent is fixed. However, if the opponent is learning and reasoning, $\phi_0$ cannot accurately represent the opponent, and thus the performance of MBOM w/o IOPs drops. By explicitly simulating the recursive reasoning process, MBOM shows more obvious performance gain against naïve learner and reasoning learner, as in Table 2. In One-on-One, the shooter has absolute dominance and more flexible strategy choices, including shooting location, shooting angle, and shooting time. Due to the limitation of gradient information, LOLA-DiCE still shows poor adaptability. Meta-PG and Meta-MAPG heavily rely on the reward signal, and thus adaptation is difficult for the two methods in this sparse reward task. MBOM w/o IOPs and MBOM achieve similar performance with fixed policy, but the performance of MBOM significantly improves if the opponent is improving as shown in Table 3, which is contributed to the recursive reasoning capability given by recursive imagination in the environment model and Bayesian mixing that quickly captures the learning of opponent.

## 5.3 COOPERATIVE TASK

MBOM could also be applied to cooperative tasks. We test the performance of MBOM on a cooperative scenario, Coin Game, which is a high-dimension expansion of the iterated prisoner dilemma

Table 2: Performance on Triangle Game

|  | Fixed Policy | Naïve Learner | Reasoning Learner |
| --- | --- | --- | --- |
| LOLA-DiCE | -22.513 (18.208) | -20.477 (18.914) | -21.554 (18.708) |
| Meta-PG | -3.777(6.034) | -6.718 (5.245) | -8.350 (11.779) |
| Meta-MAPG | -2.007 (5.639) | -5.764 (5.180) | -6.136 (9.911) |
| PPO | -13.292 (8.223) | -18.416 (8.084) | -20.508 (12.335) |
| MBOM w/o IOPs | **-1.142 (3.895)** | -3.230 (2.583) | -7.101 (16.985) |
| MBOM | -1.188 (3.871) | **-1.659 (2.817)** | **-2.746 (14.243)** |

Table 3: Performance on One-on-One

|  | Fixed Policy | Naïve Learner | Reasoning Learner |
| --- | --- | --- | --- |
| LOLA-DiCE | -0.339 (0.414) | -0.496 (0.365) | -0.632 (0.433) |
| Meta-PG | -0.165 (0.417) | -0.291( 0.371) | -0.356 (0.439) |
| Meta-MAPG | -0.112 (0.419) | -0.268 (0.365) | -0.378 (0.433) |
| PPO | -0.188 (0.274) | -0.569 (0.260) | -0.483 (0.309) |
| MBOM w/o IOPs | 0.188 (0.372) | -0.023 (0.382) | 0.113 (0.465) |
| MBOM | **0.190 (0.378)** | **0.025 (0.377)** | **0.284 (0.403)** |

The numbers in table refer to mean and standard deviation of return over 30 opponents with 5 different random seeds.

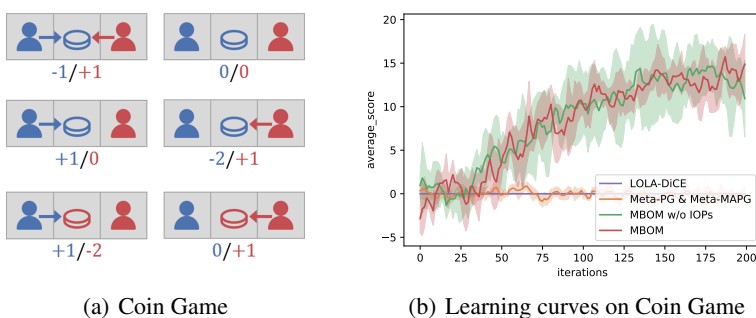

(a) Coin Game          (b) Learning curves on Coin Game

Figure 4: (a) Illustration of Coin Game; (b) Learning curves in Coin Game, which are plotted using mean and standard deviation with five runs with different random seeds.

with multi-step actions (Lerer & Peysakhovich, 2018; Foerster et al., 2018a). There are two players, red and blue, moving on a $3 \times 3$ grid field, and two types of coins, red and blue, randomly generated on the grid field. If the player moves to the position of the coin, the player collects the coin and receives a reward of $+1$. However, if the color of the collected coin is different from the player's color, the other player receives a reward of $-2$, as illustrated in Figure 4(a). The length of the game is 150 timesteps. Both agents simultaneously update their policies using the same MBOM or other baselines to maximize the sum of rewards.

The experiment results are shown in Figure 4(b). Meta-PG and Meta-MAPG degenerate to Policy Gradient for this task as there is no training set. Both learn a greedy strategy that collecting any color coin, which leads to total score of two players is zero. LOLA-DiCE learns too slow and does not learn to cooperate within 300 iterations, indicating the inefficiency of estimating the opponent gradients. MBOM w/o IOPs and MBOM learn to cooperate quickly and successfully, and MBOM shows smaller standard deviation than MBOM w/o IOPs, indicating that MBOM can also be applied to cooperative tasks without negative effect.

## 6 CONCLUSION AND FUTURE WORK

We have proposed model-based opponent modeling. MBOM employs recursive imagination and Bayesian mixing to predict and capture the learning and improvement of opponents. Empirically,

we evaluated MBOM in two competitive environments, and demonstrated MBOM adapts to learning and reasoning opponents much better than the baselines. These make MBOM a simple and effective RL method whether opponents be fixed, continuously learning, or reasoning in competitive environments. Moreover, we also verified the adaptation ability of MBOM in cooperative environments.

In MBOM, the learning agent treats all opponents as a joint opponent. If the size of the joint opponent is large, the agent will need a lot of rollouts to get an accurate best response. The cost increases dramatically with the size of the joint opponent. How to reduce the computation overhead in such scenarios will be considered in future work. Moreover, MBOM implicitly assumes that the relationship between opponents is fully cooperative. Dealing with the case where their relationship is non-cooperative is also left as future work.

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

# A  HYPERPARAMETERS

The hyperparameters of MBOM are summarized in Table 4.

Table 4: Hyper-parameters

|  |  | Triangle Game | One-on-One | Coin Game |
|---|---|---|---|---|
| PPO | policy hidden units | MLP[64,32] | LSTM[64,32] | MLP[64,32] |
|  | value hidden units | MLP[64,32] | MLP[64,32] | MLP[64,32] |
|  | activation function | ReLU | ReLU | ReLU |
|  | optimezer | Adam | Adam | Adam |
|  | learning rate | 0.001 | 0.001 | 0.001 |
|  | num. of updates | 10 | 10 | 10 |
|  | value discount factor | 0.99 | 0.99 | 0 |
|  | GAE parameter | 0.99 | 0.99 | 0 |
|  | clip parameter | 0.115 | 0.115 | 0.115 |
| Opponent model | hidden units | MLP[64,32] | MLP[64,32] | MLP[64,32] |
|  | learning rate | 0.001 | 0.001 | 0.001 |
|  | batch size | 64 | 64 | 64 |
|  | num. of updates | 10 | 10 | 10 |
| IOPs | num. of levels $M$ | 3 | 3 | 2 |
|  | learning rate | 0.005 | 0.005 | 0.005 |
|  | update times | 3 | 3 | 3 |
|  | rollout horizon | 2 | 5 | 1 |
|  | decayed factor of $\Psi$ | 0.9 | 0.9 | 0.9 |
|  | horizon of $\Psi$ | 10 | 10 | 10 |
|  | s-softmax parameter | 1 | $1.1/e$ | 1 |

# B ALPHA WEIGHTS ANALYSIS

Figure 5 visualizes $\boldsymbol{\alpha}$ of IOPs when the agent against different types of opponents, *i.e.*, fixed policy, naïve learner, and reasoning learner in Triangle Game and One-on-One. With the three opponent types, $\alpha_2$ (the weight of level-2 IOP) is remarkably higher than others, which indicates that recursive imagination of MBOM does learn a pervasive policy of the opponent by the environmental model.

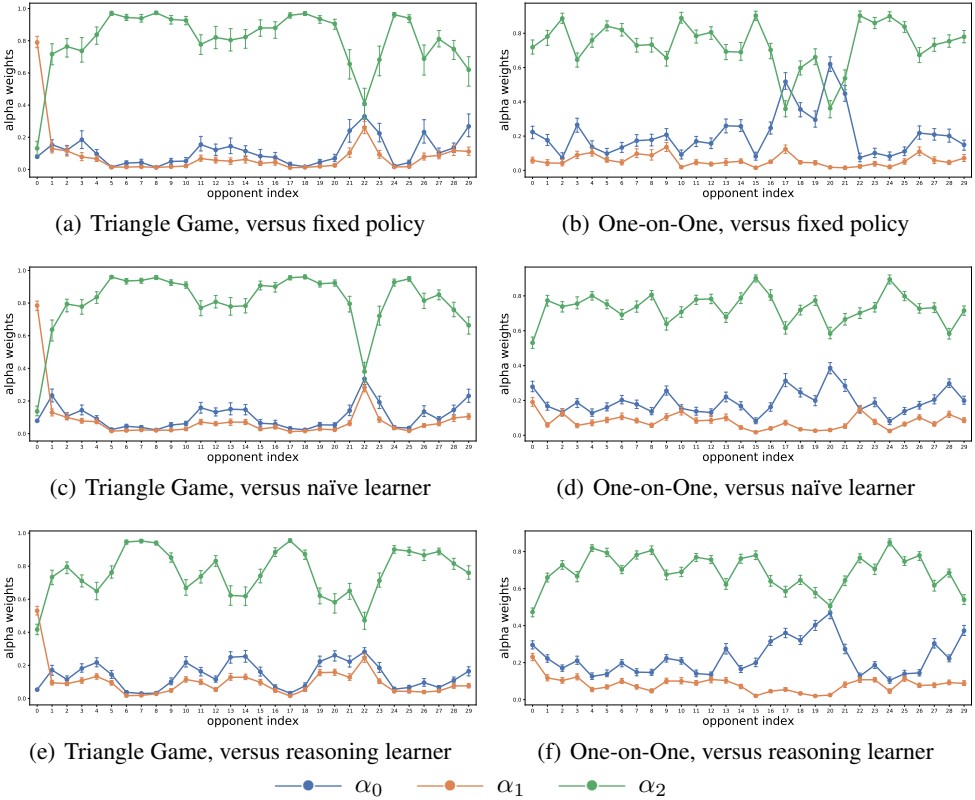

Figure 5: Visualization of $\alpha_0, \alpha_1, \alpha_2$ of MBOM, where $\alpha_i$ is the weight of level-$i$ IOP. The results are plotted using mean and standard deviation with five different random seeds (*x*-axis is opponent index).

