# OpenReview forum: "Model-Based Opponent Modeling"
_ICLR.cc/2022/Conference — ICLR 2022 Submitted_

### Official Review · Reviewer_SaHd · 2021-10-21

**Correctness:** 2
**Technical Novelty And Significance:** 3
**Empirical Novelty And Significance:** 3
**Recommendation:** 5
**Confidence:** 4

**Main Review:**

**Review**
This paper offers an interesting idea of effectively simultaenously modelling the same opponent under differing assumptions of their learning's complexity. It then used model-based predictive control with whichever opponent model is most similar to the opponents recent behavior (measured by comparing realization plans). This is a good, straight-forward, solution to the uncovered problem. Despite this, I am concerned both by the lacking of methodological details required to evaluate their empirical results, and the amount unsupported claims scattered throughout the paper.

**Major Comments**
- The set of baseline methods used are incredibly sophisticated making cross-comparisons very challenging. Simpler baselines should be included to help understand the benefit of the additional sophistication. Some easy examples are: variations on direct BR. Train a BR directly against a single opponent, mixture of the opponents, etc.. Moreover, a BR can be trained against the evaluation opponents to set an approximate upper bound on the performance these methods. Naive learner and regular LOLA can also be a baseline method, similar with other recrusive reasoning approahes discussed.
- The proposed major contribution of this work is a learning-aware learning algorithm that is independent of the other-agent's learnin algorithm. This includes many claims about various levels of required recursive reasoning being necessary to achieve desired results. I would have liked to see quantitive and qualitative experiments that analyze the inter-relation of the levels of recursive reasoning of the agent wrt the levels of recursive reasoning performed by the other-agent. This would help the reader understand how necessary and under what settings MBOM is useful. Moreover, analysis into the distribution over the IOPs can provide further insights.
- The authors liter throughout the paper unsupported claims that are often easily disproven. These distract the reader and undermine the credibility of the authors and the results herewithin. I've pointed out some of these within the Minor comments. Please ensure that claims are supported by either theoretical/empirical evidence or approriate citations.
- I am having trouble following the construction and evaluation methodology wrt to the opponent policies. It would be helpful if included, perhaps in the appendix, was more details regarding the construction of the set of opponent policies. Especially focusing on evidence that supports that these policies are both diverse and represent a good coverage of the strategic space of the game. Additionally, how each evaluated algorithm was tuned and the opponent's algorithms. Without these details it's hard to appreciate the results provided and support claims such as "out of distribution opponents" -- your evaluation opponents may in-fact be within the same distribution and there is another issue.
- If the coin game is a "high-dimension expansion" of IPD, it's confusing to me why there is an absolute failure to learn in LOLA-DiCE in Sec 5.3. IPD is the evaluation domain that is used in their work. Are you using their original implementation, or how have you evaluated that this method is working correctly? Has equal/sufficient tuning been applied to this method?
- The "Bayesian mixing" methods proposed share many similarities with previous work on purification of mixed-strategy opponents and realization plans. Appropriate citations and dicussions of these relationships is necessary to contextualize the contributions here.

**Minor Comments (not directly impacting score)**
- Abstract (and throughout), "opponents who are learning simultaneously or cpaable of reasoning" please be careful about mentioning reasoning vs recursive reasoning, and define these terms early. I.e., reasoning doesn't say anything, should only use recursive reasoning.
- Sec 1, Par 1, "interacting with diverse opponents make the enviroment nonstationary" this is not correct, stationarity of the environment is independent of the opponent. Upon abstracting the environment and other-agents into a black-box meta-environment, this meta-environment is only nonstationary if one of the components, environment or other-agent, is non-stationary (in the case of the other-agent being stochastic as mentioned here is insufficient).
- Sec 1, Par 3, "Inspired from the ... mechanics of the environment." I don't find this intuitive, humans  do not understand the reuls and mechanics of our environment. Suggestion, weaken staement, provide citations to recursive reasoning literature; esp. levels of recursive reasoning humans typically do
- Sec 1, Par 3, "Inspired from the intuition that human could anticipate ..." Awkward
- Sec 1, Par 3, "The higher-level oppoennt model reasons more deeply and thus more competitive" Awkward, and not strictly true, please justify such a claim.
- Sec 1, Par 3, "a certain-level opponnt model", undefined terminology
- Sec 2.1, Par 1, "it is a big challeng to form robust policy" Awkward
- Sec 2.1, Par 2, "One simple idea ... is encountered", worth mentioning that this requires either the agent being notified of the opponent changing or defining a function that signals opponent change -- big assumptions, the later being error prone.
- Sec 2.1, Par 2, "A better appraoch is to represent an opponent's policy with an embedding vector",  why is this a better approach? Please justify. It seems full models would be more accurate, but embeddings offer a more computationally tractable solution.
- Sec 2.1, Par 3, "Since PR2 uses the agent's.... only be applied in cooperative environments." This isn't true. Similar to this work, if assumptions are made about the game, such as zero-sum, we can infer opponent Q-values.
- Sec 2.1, Par 3, "However, these methods use ... cannot handle diverse opponents in execution." This is not generally true. The effectivness of a method's generalization to held-out opponent policies depends on how well the fixed set of agents covers the strategic landscape of the game. A point that the proposed algorithm is also vulnerable too.
- Sec 2.1, Par 4, "These meta-learning based methods require .... same learning algorithm" why does the trajectory distribution imply similarity in learning algorithm?
- Sec 2.2, Par 1, "Model-based RL allows the agent to have access to the transition function" Careful with phrasing, this isn't true. They depend on a transition function that may be learned or exact.
- Sec 4.1, Par 1, "To adapt to the learning and reasoning opponent... more deeply than the opponent" why must this be true?
- Sec 4.1, Par 2, "obtain the level-0 IOP", Awkward
- Sec 4.2, Par 1, "recursive imagination" lots of interchanging of terminiolgy between simulation/imagination/reasoning, suggest adopting and defining one nomenclature and staying consistent.
- Sec 4.2, Par 1, "a softy" ?
- Sec 4.2, Par 1, How are you calculating p(i)?
- Sec 4.2, Par 1, distinquishing where the prior work softer-softmax ends and modificaitons to work in this application begins can help the reader better understand contributions.
- Sec 5.2, Par 1, "The learning of LOLA-DiCE ... opponent quickly and effectively." This presumes you're training LOLA-DiCe against a set of opponent policies, which is not the same problem defn the solution was designed for; however, with batches of experiences carefully constructed across the opponent policies the gradient information could be effective. Especially when the gradients agree.
- Sec 5.2, Par 1, please avoid describing results as significant without corresponding statistical tests
- Sec A, Triangle mispelled in table.



**Summary Of The Paper:**

Agents learning in systems with other-agents that may be simultaneously learning presents the original agent with a dynamic learning problem. Critically, the agent does not know if the other-agents are non-learning, learning, or learning using recursive-reasoning. One such solution to this problem is to design a learning algorithm that accounts for how the other-agents may update their policy from the shared experiences, effectively eliminating the aforementioned problem. This work attempts to design such an algorithm without relying on any assumptions on the other-agent's learning algorithm. Their proposed algorithm, Model-Based Opponent Modeling (MBOM), learns parallel opponent models that are trained using various depths of recursive reasoning. Then, when played against an opponent, the agent can select the most likely opponent model (level-0: non-learning opponent, level-1: learning-opponent, level-k: k-1 level recursivly learning opponent). This algorithm assumes a centralized training and decentralized execution setting for two-player fully-observable general-sum games.

**Summary Of The Review:**

I recommend rejecting the paper in its current state. While the authors identify a problem and provide a nice and simple solution to the problem, these are undermined by experiments that are underspecified and a lack of analytical experiments into the interworkings of the algorithm.

---

> ### Author Response · Authors · 2021-11-23
> **Response to Reviewer SaHd (Part  III)**
>
> > Sec 4.2, Par 1, distinquishing where the prior work softer-softmax ends and modificaitons to work in this application begins can help the reader better understand contributions.
>
> Softer-softmax is a variant of softmax, the function form is
> $P_{y_i}=\frac{(\lambda e)^{y_i}}{\sum_j{(\lambda e)^{y_j}}}$, where $\lambda$ is a hyperparameter.
>
> > Sec 5.2, Par 1, "The learning of LOLA-DiCE ... opponent quickly and effectively." This presumes you're training LOLA-DiCe against a set of opponent policies, which is not the same problem defn the solution was designed for; however, with batches of experiences carefully constructed across the opponent policies the gradient information could be effective. Especially when the gradients agree.
>
> It is true that LOLA is not designed for such scenarios. Because it is a method that takes into account changes of the opponent's policy, so we made a fair comparison as baseline. As for whether the gradient can distinguish the opponent, I think it is similar to identifying the functions from the value of the derivative at a certain point.
>
> [1]Jakob Foerster, Richard Y Chen, Maruan Al-Shedivat, Shimon Whiteson, Pieter Abbeel, and Igor Mordatch. Learning with opponent-learning awareness. In AAMAS, 2018.
> [2] Ying Wen, Yaodong Yang, Rui Luo and Jun Wang. Modelling Bounded Rationality in Multi-Agent Interactions by Generalized Recursive Reasoning. IJCAI, 2020.
> [3] Ying Wen, Yaodong Yang, Rui Luo, Jun Wang, and Wei Pan. Probabilistic recursive reasoning for multi-agent reinforcement learning. ICLR, 2019.
> [4] Yuan, L., Fu, Z., Zhou, L., Yang, K., & Zhu, S. C. Emergence of Theory of Mind Collaboration in Multiagent Systems. arXiv:2110.00121.

---

> ### Author Response · Authors · 2021-11-23
> **Response to Reviewer SaHd (Part II)**
>
> > If the coin game is a "high-dimension expansion" of IPD, it's confusing to me why there is an absolute failure to learn in LOLA-DiCE in Sec 5.3. IPD is the evaluation domain that is used in their work. Are you using their original implementation, or how have you evaluated that this method is working correctly? Has equal/sufficient tuning been applied to this method?
>
> LOLA-DiCE doesn't provide its performance in coin game, but we can refer to LOLA's performance in coin game presented in [1]: reward got by LOLA agents begins to increase after about 500 iterations. In our experiment, we use the original code of LOLA-DiCE and its hyper-parameters, but only show the first 200 iterations. So it is reasonable to see LOLA-DiCE's "failure".
>
> In addition, we make some changes to the reward in our experiments. In LOLA paper, each agent updates its policy using its individual reward (i.e. +1 for it picking up a coin and -2 for the other picking up a wrong color coin). To test our MBOM in a fully cooperative setting, we set that 2 agent share the same reward, which is the sum of their individual rewards. This setting does not change the optimal policy (i.e. only collect its own color coin), so we believe that it wouldn't have a significant impact on LOLA's performance.
>
> > The "Bayesian mixing" methods proposed share many similarities with previous work on purification of mixed-strategy opponents and realization plans. Appropriate citations and dicussions of these relationships is necessary to contextualize the contributions here.
>
> We found a similar mixing method in [2], and have added the citation in the revision.
>
> > Sec 2.1, Par 2, "A better appraoch is to represent an opponent's policy with an embedding vector", why is this a better approach? Please justify. It seems full models would be more accurate, but embeddings offer a more computationally tractable solution.
>
> We admit that "better" is not accurate. Maybe "more computationally tractable" is more precise. There exists trade-off between model accuracy and computational effciency, so we can't tell which is "better".
>
> > Sec 2.1, Par 3, "Since PR2 uses the agent's.... only be applied in cooperative environments." This isn't true. Similar to this work, if assumptions are made about the game, such as zero-sum, we can infer opponent Q-values.
>
> It's true that we can change $Q^i$ to $-Q^i$ to make PR2 work in zero-sum game. But what we mean here is that PR2 cannot be applied on non-cooperative games without making any change. We'll make this statement more clear.
>
> > Sec 2.1, Par 3, "However, these methods use ... cannot handle diverse opponents in execution." This is not generally true. The effectivness of a method's generalization to held-out opponent policies depends on how well the fixed set of agents covers the strategic landscape of the game. A point that the proposed algorithm is also vulnerable too.
>
> "These methods" [3] [4] aim to train a group of agents to develop effective collaboration. These agents will take actions according to the protocol they learned in centralized training, so they cannot cooperate with other agents who follow different protocols, which "diverse opponents" means here.
>
> > Sec 2.1, Par 4, "These meta-learning based methods require .... same learning algorithm" why does the trajectory distribution imply similarity in learning algorithm?
>
> In the setting of continuous adaptation, trajectories used for adaptation include a chain of episodes, and opponents are allowed to update their policies between episodes. So trajectory distribution includes intra-episode transition $(s, a, s', r)$ distribution and inter-episode policy change. When opponents use the same learning algorithm in meta-training and meta-test, the distribution of trajectories will match across training and test.
>
> > Sec 4.1, Par 1, "To adapt to the learning and reasoning opponent... more deeply than the opponent" why must this be true?
>
> From the view of our agent, the learning opponents' update or reasoning opponents' recursive reasoning make the environment non-stationary, which means naive single RL algorithm wouldn't work well. Predicting opponents' policies or reasoning more deeply is a sufficient condition for better performance, compared with naive RL.
>
> > Sec 4.2, Par 1, How are you calculating p(i)?
>
> $p(i)$ is just $p(m)$ with a different index. As mentioned in Sec 4.2, Par 1, it is the moving average of $p(m|a_o)$:
> $$
> {p}(m)=\frac{1}{H}\sum_{l=t-H}^{t-1}p(m|a_{o}^{l})
> $$

---

> ### Author Response · Authors · 2021-11-23
> **Response to Reviewer SaHd (Part I)**
>
> Thanks for your valuable comments. In the following, we provide a detailed explanation of your concerns.
>
> > The set of baseline methods used are incredibly sophisticated making cross-comparisons very challenging. Simpler baselines should be included to help understand the benefit of the additional sophistication. Some easy examples are: variations on direct BR. Train a BR directly against a single opponent, mixture of the opponents, etc.. Moreover, a BR can be trained against the evaluation opponents to set an approximate upper bound on the performance these methods. Naive learner and regular LOLA can also be a baseline method, similar with other recrusive reasoning approahes discussed.
>
> We have added naive learner (PPO) as a baseline in the revision (Figure 3) to make the comparison more clear. As expected, PPO does not perform well. However, we think LOLA is unnecessary to be a baseline, because we already used LOLA-DiCE as a baseline, which is an improvement over LOLA by producing the correct higher order gradient. Morover, we don't think that BR can be a baseline, because it is necessary to know the type of opponents (i.e. fixed, learning or reasoning) in advance so that we could build and learn a best response policy, which does not conform to our assumption that our agent do not know which type the opponents are. Using BR as a upper bound might be an option. However, it incurs huge training cost, since for the opponet of naive learner and reasoning learner, BR has to be retrained after the opponent updates its policy.
>
> > The proposed major contribution of this work is a learning-aware learning algorithm that is independent of the other-agent's learnin algorithm. This includes many claims about various levels of required recursive reasoning being necessary to achieve desired results. I would have liked to see quantitive and qualitative experiments that analyze the inter-relation of the levels of recursive reasoning of the agent wrt the levels of recursive reasoning performed by the other-agent. This would help the reader understand how necessary and under what settings MBOM is useful. Moreover, analysis into the distribution over the IOPs can provide further insights.
>
> We have added the analysis on mixing weight in Appendix (Figure 5). Although there is no tight level matching, the results also show that higher-level IOPs of the agent do play an important role in the interaction.
>
> > The authors liter throughout the paper unsupported claims that are often easily disproven. These distract the reader and undermine the credibility of the authors and the results herewithin. I've pointed out some of these within the Minor comments. Please ensure that claims are supported by either theoretical/empirical evidence or approriate citations.
>
> For Minor comments, we answer questions in detail in the following, and those not answered are corrected in the revision.
>
> > I am having trouble following the construction and evaluation methodology wrt to the opponent policies. It would be helpful if included, perhaps in the appendix, was more details regarding the construction of the set of opponent policies. Especially focusing on evidence that supports that these policies are both diverse and represent a good coverage of the strategic space of the game. Additionally, how each evaluated algorithm was tuned and the opponent's algorithms. Without these details it's hard to appreciate the results provided and support claims such as "out of distribution opponents" -- your evaluation opponents may in-fact be within the same distribution and there is another issue.
>
> Regarding the details of the experiment, "To increase the diversity of the opponents..." is mentioned in Sec 5.1, Par 7, which is not really detailed enough. For the Triangle Game, we trained the opponents set with a modified reward, as shown in the table below, so that we could get the opponent that commuting between T1 and T2. Other types of opponents, such as hovering around a landmark, commuting between 2 landmarks, or rotating among 3 landmarks, are obtained in a similar way. For One-on-One, we set a barrier in front of the goal (invisible, but can block the ball) and only keep a gap so that the ball can enter the goal. We trained opponents with different shooting tendencies in environments with gaps in different positions.
>
> |     |  F  | T1  | T2  | T3  |
> |  ----  | ----  | ----  | ----  | ----  |
> |  F   |  -0.1/-0.1  |  -0.1/1 | -0.1/1  | -0.1/-0.1  |
> |  T1   |  1/-0.1  | -0.5/1  | 1/-0.5  | 1/-0.1  |
> |  T2   |  1/-0.1  | 1/-0.5  | -0.5/1  | 1/-0.1  |
> |  T3   |  -0.1/-0.1  | -0.1/1  | -0.1/1  | -0.1/-0.1  |

---

> > ### Comment · Reviewer_SaHd · 2021-11-24
> > **Reviewer Response**
> >
> > **Baselines**
> > Thank you for adding an additional baseline. I think the major point of my comment here was missed. The thing that I am looking for to help understand the method is additional analysis of the results. The suggested baselines were simply to offer suggestions as means to glean more insights into the success of the method.
> >
> > **Opponent Diversity**
> > The reward shaping details does help understand the diversity of the agents. However, the shaped rewards doesn't necessarily mean that the policies themselves are behaving differently.
> >
> > **Minor Comments**
> >  - "When opponents use the same learning algorithm in meta-training and meta-test, the distribution of trajectories will match across training and test." This is again the same claim that I don't believe is supported/known.
> >  - "Predicting opponents' policies or reasoning more deeply is a sufficient condition for better performance, compared with naive RL." If you predict the policy I'm still not sure why the comment about reasoning more deeply is necessary, nor is the meaning of that statement clear.
> >  - Various: Some of the minor comments are for suggestions to improve the clarity of the text, and not necessarily to inform the reviewers.

---

### Official Review · Reviewer_8g3Q · 2021-11-01

**Correctness:** 3
**Technical Novelty And Significance:** 3
**Empirical Novelty And Significance:** 3
**Recommendation:** 5
**Confidence:** 4

**Main Review:**

The paper address a significant problem of multi-agent reinforcement learning in the setting when agents are 1) heterogeneous 2) learn/reason about each other and adapt their policies based on observations of other agents' behavior. In the general discussion, the paper covers relevant related work, and discusses challenges and limitations. Empirical evaluations conclude both quantitative evaluation of performance and insights on the possible mechanisms and causes of observed performance patterns, and justifications of the proposed algorithm. This research can potentially become a significant and influential contribution to reinforcement learning community.

That said, I have several reservations about the paper and would like to see if they can be addressed in a revision.

1) The empirical evaluation is not well aligned with related work. It compares MOBM with other baselines on three domains, one of them is collaborative and two are competitive. It seems that none of these domains appears in the same form in the cited work to which the new algorithm compared. Since this paper claims that SOTA, I would anticipate that the comparison is performed on domains on which the baselines claim SOTA, in addition to the new domains introduced or adapted in the paper.

2) The main idea of the paper is in representation learning for recursive epistemic reasoning. However, convergence of such reasoning, and soundness of Bayesian mixing of such methods is not theoretically analyzed. This is not obvious at all that recursive epistemic reasoning (theory of mind) works in competitive domains, and what are the restrictions/conditions under which it is sound.
  a) First, recursive epistemic reasoning is applied to competitive tasks. It is easy to show that it does not always converge. Consider Bob and Alice and two bars settings. Bob and Alice slightly prefer bar A, but both what to avoid each other. This is a competitive setting. Let us reason with Alice. On level 0, Alice chooses bar A. On level 1, Alice realizes that Bob chooses bar A too, so she chooses bar B. On level 2, Alice thinks about Bob choosing bar B on level 1 so she chooses bar A. On level 3, she would switch to bar A again, and so on. No convergence. By making Bob's and Alice's policies probabilitistic the analysis can be made more interesting, but still not converging under certain conditions to any equilibrium. This is not an edge case but a general property of recursive reasoning, which does not seem to be addressed in the paper.
 b) Second, recursive reasoning is built upon the best imagined action of the opponent. This is a problematic setting, related to Newcomb's paradox. Consider, again Bob and Alice, in a setting in which Alice has equal preferences regarding the bars and attributes reward 1 to avoiding Bob and 0 to meeting him. Bob chooses the first bar in 55% of cases; Alice employs a single level of epistemic
reasoning. It is easy to see that the optimal policy for Alice is to always go to the second bar for the expected reward or 0.55.
However, the policy inferred using conditioning on the best anticipated move of the opponent is  is for Alice to choose the second bar with
probability of 0.55, for the expected reward of 1! This is because Alice's decision is (erroneously) conditioned on Bob's
anticipated choice of a bar, and hence Alice pretends that she can always choose the other bar (which she cannot). This issue also seems to be omitted in the paper. The agent behavior may seen reasonable in the complicated settings of the proposed domains, but in general I suspect that the algorithm is not sound.

Based on the above, I would appreciate a formal theoretical treatment and an empirical evaluation to be improved in a revised version of the paper.



**Summary Of The Paper:**

The paper proposes MBOM, a scheme for multi-agent reinforcement learning, which flexibly adapts to opponents with different or unknown learning abilities by relying on recursive reasoning.  The paper evaluates the algorithm on two competitive environment, empirically comparing to other baseline and state-of-the-art schemes.

**Summary Of The Review:**

An interesting line of research but theoretical analysis and empirical evaluation should be improved in a revised version.

---

> ### Author Response · Authors · 2021-11-23
> **Response to Reviewer 8g3Q**
>
> Thanks for your comments. Below is our detailed explanation of your concerns.
>
> >1. The comparison is performed on domains on which the baselines claim SOTA
>
> One of the major advantages of MBOM is that it fits both zero-sum and fully cooperative settings. Current experimental scenarios are actually used to support this claim. Moreover, the coin game is indeed from the paper of LOLA. Unlike existing work, we envision MBOM is a practical solution for complex scenarios. Therefore, we use One-on-One which is a much more complex zero-sum game than those used in the papers of baselines. The superior performance of MBOM in these scenarios adequately supports our claims.
>
> > 2.a. The convergence of recursive reasoning.
>
> First, we need clarify that our aim is not to enable the agent to reach an equillibrium or convergence with the opponent in competitive scenarios. Instead, our goal is to enable the agent to perform better against the different types of opponents.  Actually, there is no convergence of recursive reasoning, but the lack of convergence does not limit MBOM. During recursive imagination, we obtain the models (IOPs) of opponents at different levels. Then in Bayesian Mixing, we mix the IOPs to obtain a more accurate estimate of the improving opponent policy, which could be seen as the opponent policy with the real level. That is to say, no matter whether the IOPs in recursive imagination converge or not, the agent could infer the real-level opponent policy by Bayesian Mixing and adapt to it.
>
> > 2.b. Newcomb's paradox
>
> The problem you proposed is an inherent issue in the setting where the opponent takes the stochastic policy. It is easy to solve this problem by conditioning the agent policy on the action distribution of the imaged opponent policy, e.g. the mean and the std, rather than the sampled action of imaged opponent policy. In fact, we also input the action distribution of imaged opponent policy into the policy network in the experiments. As you said, in complicated environments, this issue would not greatly impact the performance, especially when the opponent policy is nearly deterministic. Even in the case where the policy of Bob is nearly uniform, the expected return of inputting sampled action is $0.55\*0.55+0.45\*0.45=0.505 $​​, which is slightly lower than $0.55$.

---

### Official Review · Reviewer_KD9g · 2021-11-02

**Correctness:** 3
**Technical Novelty And Significance:** 3
**Empirical Novelty And Significance:** 3
**Recommendation:** 5
**Confidence:** 4

**Main Review:**

Major comments:

1. When estimating the next level opponent, at time step t, the opponent's action is selected from the best action using the rollout with highest value, estimated by expanding k steps with the dynamics model. Below Algorithm 1, "With larger k, the rollout has longer planning horizon, and thus could evaluate the action a^{o*} more accurately" is problematic by noting that the rollout is generated by interacting with a learned dynamics model, which suffers uncertain dynamics error. Moreover, for zero-sum or cooperative setting, the opponent's value is estimated by directly using the negative self agent's value or the self agent's value. This is also questionable since for multi-agent settings, the problem is not an MDP again from the perspective of one agent. So, the value estimated from the self agent's observation conditions on only partial states without knowing the opponent's information. Based on this, the value is not exchangeable. Overall, I think the recursive reasoning part might rise many uncertainty issues and suffer approximation errors at least from the above two ways, and these uncertainty and error can be accumulated for higher level opponent reasoning. Unless the dynamics model error and value estimation error can be analyzed theoretically to guarantee the higher level opponent's value is also higher from the opponent's view, I am wondering whether the proposed recursive reasoning method can consistently generate stronger opponents or not.

2. What is the case when the problem is not zero-sum? Also, when there are more than one opponents, would the recursive reasoning becomes more complicated, because when use a model to generate rollouts, the best response rollout should be selected by considering other opponents' actions?

3. In the experiments, two-player games are considered. I think the original LOLA and a plain RL baseline without any reasoning should also be compared as references. Moreover, as the number of levels of IOPs and the length of rollouts k are important factors in the proposed method, these hyper-parameters should be studied with sufficient empirical results, which are absent from the current contexts. Most importantly, I think experimental results that can visualize the difference and improvements along the opponent models from lower level to higher level should be crucial to support the effectiveness of the proposed recursive reasoning. However, these results are not considered in the current experiment section.

Overall, this paper proposes an interesting recursive way for modeling the opponent's policy, while there exists many issues to be solved in revisions.




**Summary Of The Paper:**

This paper proposes a model-based opponent recursive reasoning method for MARL problem. The recursive reasoning takes multiple levels of estimation to generate a number of opponents. The first level opponent is obtained by supervised learning from the true opponent actions. Then, with a learned dynamics model, the level-0 opponent interacts with self agent and the dynamics model to generate trajectories, from which by selecting one best response rollout, a better action at the current time step for the opponent is kept and used to estimate the next level opponent, by maximum log-likelihood again. After obtaining a number of levels of opponents, a mixture opponent is learned by softer-softmax weights. Experiments are conducted on multi-agent particle environment and google's football environment, and LOLA-DiCE and Meta-PG are compared.

**Summary Of The Review:**

An interesting model-based opponent modeling method in MARL while approximation errors in recursive reasoning and more explanatory experimental results should be considered.

---

> ### Author Response · Authors · 2021-11-23
> **Response to Reviewer KD9g**
>
> Thank you for acknowledging our novel contributions as well as raising valuable questions.
>
> > The dynamics model error and value estimation error
>
> In our settings, the agent and the opponent could obtain the global state $s$. In zero-sum games, $V = E_{\pi}E_{\pi^o}\sum_{t=0}^{\infty} \gamma^{t}r_t$, $V^o = E_{\pi^o}E_{\pi}\sum_{t=0}^{\infty} \gamma^{t}r_t^o$, thus $V + V^o = E_{\pi,\pi^o}\sum_{t=0}^{\infty} \gamma^{t}(r_t + r_t^o) = 0$. So we estimate the opponent state value $V^o(s)$ as $-V(s)$​. This also applies to fully cooperative settings.
>
>
> Thank you for pointing out the incorrect statement, "With larger k, the rollout has longer planning horizon, and thus could evaluate the action a^{o*} more accurately." We have reword the statement.  Admittedly, yes, there is a tradeoff between the model error and the accuracy of best response. Currently, in the experiments, the model is trained on a large dataset, and we empirically found the model is very accurate such that the model error is negligible for $k$ we chose in the experiments. The accuracy of the best response not only depends on the model error, but also the number of Monte Carlo samples. Let us say we could have enough samples to get an accuracy best response given the model. Then, the model error is the only source of the inaccuracy of the best response. In such a way, the theoretical analysis in MBPO [1] directly applies. **However, we do not think the theoretical analysis can be easily extended further to recursive reasoning, since the rollout for the next level uses the agent policy conditioned the opponent model which is fintuned by the best response of the current level.** A thorought theoretical study on this can be a brand new paper. That said, our paper actually brings up many interesting problems to to solved in future work.
>
> > What is the case when the problem is not zero-sum? Also, when there are more than one opponents, would the recursive reasoning becomes more complicated?
>
> As discussed in the section of conclusion and future work, Currently, MBOM only handles zero-sum or fully cooperative settings. When there are more than one opponent, MBOM treats all opponents as a joint opponent. The growth of action dimension requires more Monte Carlo samples to obtain an accurate best response of the joint opponents. How to make MBOM generalizable to general-sum setting needs a thorough study and is left as future work.
>
> >  LOLA, a plain RL baseline, the ablation of $k$...
>
> **We have added the plain RL (PPO) in the experiments as in the revision.** As expected, it does not perform well. We do not think LOLA is necessary to be a baseline, because we already used LOLA-DiCE as a baseline, which is an improvement over LOLA by producing the correct higher order gradient. For the ablation of hyperparameter $k$, as discussed above, many existing works have studied it theoretically and empirically, which is also not our focus, and we omit it in our paper. For the effect of different reasoning levels, we do not have enough time during rebuttal to rerun the experiments to show the improvement of each level. But we have added a figure in Appendix in the revision to visualize $\boldsymbol{\alpha}$ of IOPs when the agent against different types of opponents. With the three opponent types, $\alpha_2$ (the weight of level-2 IOP) is remarkably higher than others, which indicates that recursive imagination of MBOM does learn a pervasive policy of the opponent by the environmental model.
>
> [1] Janner et al., When to Trust Your Model: Model-Based Policy Optimization

---

### Official Review · Reviewer_3i5i · 2021-11-04

**Correctness:** 2
**Technical Novelty And Significance:** 3
**Empirical Novelty And Significance:** 2
**Recommendation:** 3
**Confidence:** 3

**Main Review:**

The method proposed is elegant and seems promising, but I have a number of questions that require clarifications:
1. the method is designed to work with multiple opponents that are treated as a joint opponent.  In both and the Triangle Game and the One-on-One game e there are only two agents.  This raises the issue of what happens in situation when there is more than one opponent.  It would have been useful to have at least one application with multiple agents to understand how well the proposed algorithm addresses the multi-agent opponent modeling as claimed.
.2. The two games used for the experimental work seem to have sequential and not simultaneous moves, and the same applies to the cooperative game. It would have been useful to address the issue in the description.  Again, does the algorithm expect a sequential move game?
3. Not much is said on the opponent types that are assumed, specifically the difference between the naive learner and the reasoning learner.  This is important because the readers need to know how to decide if an opponent is "learnable".   Also, it is not clear if the agent knows the type of opponent (fixed policy, etc) or not.
4. The computational costs are mentioned as being high, but there is no indication, for instance, of the computing time taken by the examples presented in the paper. The problem of reducing the computational complexity is listed as future work.
5. There are graphs that show performance results for the games, but it is not clear how performance is measured.  Some explanations would help. The performance tables (Table 2 and Table 3) show numerical performance values but do not say what the values represent.
6. The placement of tables and figures is not ideal. They are referenced very far form where they show up in the paper, making the reading harder.
7. The paper does not include any theoretical results, and, given the comments made above, it is hard to understand when the method will work and how well it will work.

**Summary Of The Paper:**

The paper addresses the problem of learning an opponent model in a multi-agent environment. What makes learning hard is that the opponent can be learning at the same time. The paper proposes an algorithm, called MBOM, that operates by simulating the reasoning process of the agent in the environment and considering different types of opponents. The process is recursive and uses the environment model to predict the policy improvement of the opponents. The process generates improved models of the opponent, that are then used with Bayesian mixing to produce an estimate of the opponents policy. The MBOM algorithm is tested on two games, the Triangle Game and the One-on-one two player game. Different algorithms are used and their performance is compared with MBOM.

**Summary Of The Review:**

The paper presents a promising method for an agent to learn opponent models in a multi-agent environment. The method is compared empirically with similar algorithms which are outperformed.  Unfortunately, the paper does not provide theoretical guarantees, lacks details in some of the explanations, as listed above, and shows only examples with two agents, an agent and an opponent.  In conclusion, the method seem promising and well thought out,  but it is hard to be understand what are the requirements to make the method applicable in practice.

---

> ### Author Response · Authors · 2021-11-23
> **Response to Reviewer 3i5i**
>
> Thanks for your comments. As follows, we address your concerns in detail.
> > The method is designed to work with multiple opponents that are treated as a joint opponent. In both and the Triangle Game and the One-on-One game there are only two agents. This raises the issue of what happens in situation when there is more than one opponent. It would have been useful to have at least one application with multiple agents to understand how well the proposed algorithm addresses the multi-agent opponent modeling as claimed. 2. The two games used for the experimental work seem to have sequential and not simultaneous moves, and the same applies to the cooperative game. It would have been useful to address the issue in the description. Again, does the algorithm expect a sequential move game?
>
> As explicitly discussed in the section of Conclusion and Future Work, we admit that the size of the joint opponent exponentially increases the computation cost to obtain an accuracte best response, and how to reduce the cost and empirically show the performance of MBOM is left as future work due to the limited compute resources.
>
> All the games in this paper are simultaneous moves, and we mentioned it in Section 3, "All agents interact with the environment simultaneously without communication."
>
> > Not much is said on the opponent types that are assumed, specifically the difference between the naive learner and the reasoning learner. This is important because the readers need to know how to decide if an opponent is "learnable". Also, it is not clear if the agent knows the type of opponent (fixed policy, etc) or not.
>
> **MBOM does not assume the agent knows the opponent type**. These types are used for the agent to play against so as to verify the effectiveness of MBOM. As discussed in the last paragraph of Section 5.1, the naive learner just updates its policy using PPO during test, while the reasoning learner learns a model to predict the action of the agent and a policy conditioned on the predicted action of the agent, both of which are updated during test. MBOM does not require prior knowledge such as whether the opponent is learnable or not, and the opponent type is not visible for the agent. In other words, MBOM only makes inference from the past behavior of the opponent.
>
> > The computational costs are mentioned as being high, but there is no indication, for instance, of the computing time taken by the examples presented in the paper. The problem of reducing the computational complexity is listed as future work.
>
> We tested MBOM against the fixed policy opponent in One-on-One for 100 episodes. The results are shown below. The computing time depends on rollout horizon $k$ and the size of the opponent's action space. The action space of the opponent in this game is 11.​
>
> |  k   | 0  | 1  | 2  | 3  | 4  | 5  |
> |  ----  | ----  | ----  | ----  | ----  | ----  | ----  |
> | computing time (seconds per episode) | 4.7 | 13.3 | 15.6 | 18.7 | 24.4 | 52.3 |
>
> > There are graphs that show performance results for the games, but it is not clear how performance is measured. Some explanations would help. The performance tables (Table 2 and Table 3) show numerical performance values but do not say what the values represent.
>
> The numbers in the performance tables refer to  mean and standard deviation of returns over 30 opponents with five different random seeds. We have added an explanation for Table 2 and Table 3 in the revision.
>
> > The placement of tables and figures is not ideal. They are referenced very far form where they show up in the paper, making the reading harder.
>
> We will correct this.
>
> > The paper does not include any theoretical results, and, given the comments made above, it is hard to understand when the method will work and how well it will work.
>
> The motivation of MBOM is described in Section 4, which gives the intuition behind MBOM. As MBOM does not have any assumptions on the opponent types (we think this is one of major contributions of this paper), it is extremely hard to provide a theoretical guarantee on the performance.

---

> > ### Comment · Reviewer_3i5i · 2021-11-28
> > **Thanks for your answers and clarifications**
> >
> > Thanks for your answers and clarifications.  Thanks for sharing the computation times against the fixed policy opponent. Depending on the problem, the computation times might be too high, and, given the increase with k, they will be significantly higher with multiple opponents. The approach proposed is interesting and addressed an open problem, but I am not fully convinced it will work well in other application domains.  I am also wondering how well the approach will work in the case of multiple opponents, since in the experiments there is only one opponent.  I understand that it is difficult to provide theoretical results when there are no assumptions about the opponent, but that is the reason why other approaches make assumptions on the opponent types and can then prove theoretical results.  With no theoretical results, many more experiments in many domains with different properties and different number of opponents are needed to be convincing.

---

### Decision · Program_Chairs · 2022-01-20

**Decision:**

Reject

**Comment:**

This paper tackles the challenging problem of learning against an opponent that may or may not be simultaneously learning as well. The key contribution of this paper is a learning algorithm that accounts for how the opponents may update their policies from past interactions. The proposed algorithm, MBOM, relies on the environment model to model a hierarchy of opponents using different depths of recursive reasoning (from non-learning agents to deep recursive agents). It is agreed that this papers studies an important problem and shows promise. However, the current results aren't convincing enough. In particular, since there is no theoretical analysis, more empirical validation of the method is expected. The current experiments only considers a single opponent, and it is unclear how well the method works given accumulated errors through the recursion. Future submissions would benefit from additional empirical analysis (e.g., ablations) to help understand when and why MBOM works.